# Cell Penetrating Peptide Enhances the Aphidicidal Activity of Spider Venom-Derived Neurotoxin

**DOI:** 10.3390/toxins16080358

**Published:** 2024-08-14

**Authors:** Wenxian Wu, Abid Ali, Jinbo Shen, Maozhi Ren, Yi Cai, Limei He

**Affiliations:** 1Institute of Urban Agriculture, Chinese Academy of Agricultural Sciences, Chengdu Agricultural Science and Technology Center, Chengdu 610000, China; wuwenxian@caas.cn (W.W.); renmaozhi01@caas.cn (M.R.); 2Department of Entomology, University of Agriculture Faisalabad, Faisalabad 38040, Pakistan; abid_ento74@yahoo.com; 3College of Life Science, Shenyang Normal University, Shenyang 110034, China; 4State Key Laboratory of Subtropical Silviculture, Zhejiang A&F University, Hangzhou 310000, China; jshen@zafu.edu.cn; 5College of Life Science, Sichuan Agricultural University, Ya’an 625000, China

**Keywords:** insecticidal neurotoxin, cell penetrating peptide, fusion protein, aphicide, multi-copy expression

## Abstract

HxTx-Hv1h, a neurotoxic peptide derived from spider venom, has been developed for use in commercial biopesticide formulations. Cell Penetrating Peptides (CPPs) are short peptides that facilitate the translocation of various biomolecules across cellular membranes. Here, we evaluated the aphidicidal efficacy of a conjugated peptide, HxTx-Hv1h/CPP-1838, created by fusing HxTx-Hv1h with CPP-1838. Additionally, we aimed to establish a robust recombinant expression system for HxTx-Hv1h/CPP-1838. We successfully achieved the secretory production of HxTx-Hv1h, its fusion with *Galanthus nivalis* agglutinin (GNA) forming HxTx-Hv1h/GNA and HxTx-Hv1h/CPP-1838 in yeast. Purified HxTx-Hv1h exhibited contact toxicity against *Megoura crassicauda*, with a 48 h median lethal concentration (LC_50_) of 860.5 μg/mL. Fusion with GNA or CPP-1838 significantly enhanced its aphidicidal potency, reducing the LC_50_ to 683.5 μg/mL and 465.2 μg/mL, respectively. The aphidicidal efficacy was further improved with the addition of surfactant, decreasing the LC_50_ of HxTx-Hv1h/CPP-1838 to 66.7 μg/mL—over four times lower compared to HxTx-Hv1h alone. Furthermore, we engineered HxTx-Hv1h/CPP-1838 multi-copy expression vectors utilizing the *Bgl*Brick assembly method and achieved high-level recombinant production in laboratory-scale fermentation. This study is the first to document a CPP fusion strategy that enhances the transdermal aphidicidal activity of a natural toxin like HxTx-Hv1h and opens up the possibility of exploring the recombinant production of HxTx-Hv1h/CPP-1838 for potential applications.

## 1. Introduction

Herbivorous insects significantly impair global agricultural production, contributing to an estimated 10–20% loss in grain yield annually [1]. Recent shifts in climate and other environmental factors have intensified the frequency and scale of pest outbreaks, posing a critical threat to global food security [2]. Currently, chemical insecticides are the predominant method for pest control. However, prolonged and improper use of these chemicals has led to severe ecological and environmental repercussions. Beneficial organisms such as bees, birds, and aquatic invertebrates are exposed to pesticide toxicity either directly or indirectly through spray drift or leaching [3]. The persistence of highly toxic and residual chemical pesticides in ecosystems can lead to bioaccumulation, causing irreversible damage to both human health and the environment [4]. Additionally, the irrational long-term application of chemical pesticides has accelerated the development of insecticide resistance among pest populations [5]. Biopesticides are considered safer and more sustainable alternatives to chemical pesticides due to their benign nature and environmental compatibility [6]. For example, the bacterium *Bacillus thuringiensis* (Bt), which accounts for half of all biopesticide usage, produces Cry proteins that bind specifically to midgut receptors in pests, causing lethal intestinal perforations within days [7]. Moreover, Bt does not affect mammals, as Cry protein receptors are absent in the mammalian gut. However, the insecticidal range of Bt is limited to specific lepidopteran pests and is ineffective against others such as aphids and whiteflies, which possess piercing and sucking mouthparts [8].

Venomics research has revealed that arthropod predators such as spiders and scorpions possess a diverse array of neurotoxic insecticidal peptides. These peptides are typically short (2.5–5 kDa), rich in disulfide bonds, and characterized by an evolutionarily conserved inhibitor cystine knot (ICK) motif [9,10]. The ICK motif provides structural stability, making these peptides resistant to chemical, thermal, and proteolytic degradation in the insect gut and hemolymph [11]. This robustness renders neurotoxic peptides promising candidates for novel biopesticide formulations. A notable example are the Spear^®^ series bioinsecticides produced by Vestaron, which utilize ω/κ-HxTx-Hv1h (hereafter referred to as HxTx-Hv1h), a peptide derived from the venom of the Australian funnel-web spider *Hadronyche versuta*, as their primary active component [12,13]. HxTx-Hv1h is a member of the hexatoxin superfamily subgroup ω-hexatoxin-1, which also includes ω-hexatoxin-Hv1a and κ-hexatoxin-Hv1c. All three toxins are not only effective through injection across a broad spectrum of pests, but are also non-toxic to mammals and beneficial insects, aligning with integrated pest management (IPM) strategies [14,15,16,17]. Despite their varying amino acid sequences, the three-dimensional structures of these toxins are remarkably similar. Furthermore, HxTx-Hv1h is considered a hybrid toxin because it shares crucial amino acid residues with both ω-hexatoxin-Hv1a and κ-hexatoxin-Hv1c [18]. Meanwhile, HxTx-Hv1h synergistically targets insect voltage-gated calcium (Ca_v_) and potassium channels (K_Ca_), thereby integrating the insecticidal mechanisms of ω-hexatoxin-Hv1a and κ-hexatoxin-Hv1c [18,19,20]. Recent studies have demonstrated their potentiation of insect nicotinic acetylcholine receptors, providing a basis for their selective activity [21].

Arthropod predators, which deliver venom directly to their prey, typically produce toxins not naturally selected for oral or contact toxicity. Consequently, venom-derived peptides often exhibit higher oral or contact LC_50_ values compared to injection, due to barriers such as the midgut epithelial or epidermal cell layers in insects. To overcome these challenges, research has focused on effective transport carriers like the snowdrop lectin *Galanthus nivalis agglutinin* (GNA) and the development of delivery systems that enhance the bioavailability of neurotoxic peptides [12,13,16,22,23]. Bonning et al. used the capsid protein of an insect-borne plant virus to deliver a neurotoxic peptide into the hemocoel of aphids via fusion expression, rendering insect neurotoxins orally toxic [24]. Additionally, the use of certain adjuvants has been shown to facilitate peptide transport across insect cuticles, thereby improving topical and residual efficacy [25].

Another strategy for enhancing bioavailability involves conjugating molecular cargos with cell-penetrating peptides (CPPs), which are small peptides known for their ability to facilitate intracellular delivery [26]. Since the discovery that the positively charged section between amino acids 47–57 on the TAT transactivation protein of HIV can effectively penetrate the plasma membrane, numerous CPPs have been identified. These peptides exhibit diverse structural features and modes of penetration [27,28]. The delivery efficacy of CPPs is profoundly influenced by peptide sequence, cargo type, and target cells [29,30]. Ramaker et al. assessed the capacity of 474 CPPs to transport model cargos into HeLa cells using a uniform uptake assay, identifying 20 CPP sequences with significantly enhanced delivery efficiencies, including CPP-1838 [30]. Further research by Darif et al. explored the utility of CPPs as insecticide enhancers, characterizing the uptake and penetration mechanisms of four CPPs into two insect cell lines and dissected midgut tissues [31]. Their findings demonstrated variability in uptake among different CPPs, with CPP-1838 showing exceptional proficiency for cell penetration via both diffusion and endocytosis [31]. These attributes highlight CPP-1838’s exceptional cell-penetrating capability, underscoring its potential as an effective biopesticide enhancer due to its efficient delivery mechanisms.

A significant challenge in the commercialization of venom-derived insecticidal peptides lies in achieving sustainable yields through recombinant expression. Yeasts such as *Kluyveromyces lactis* and *Pichia pastoris* have emerged as effective platforms for large-scale recombinant protein synthesis. Unlike prokaryotic systems, these yeasts possess sophisticated secretion systems that facilitate the extracellular release of substantial quantities of recombinant proteins, thereby streamlining the downstream purification process. Furthermore, the eukaryotic endoplasmic reticulum within yeasts provides an optimized environment for the proper folding of small, disulfide-rich proteins into their biologically active conformations [32]. To augment recombinant protein yields in *P. pastoris*, strategies typically involve the amplification of gene dosage by inserting multiple copies of the target gene into the host genome. This can be done by either screening for strains with multiple gene copies using antibiotic resistance or by creating recombinant vectors with multiple genes for genome integration, which leads to transformants with more gene copies [33,34]. Generally, a positive correlation is observed between gene copy number and protein expression levels, although there exists an optimal threshold beyond which additional increments may inversely affect expression [35].

In this study, we reported the successful production of recombinant neurotoxic peptides: HxTx-Hv1h alone, as well as its conjugates HxTx-Hv1h/GNA and HxTx-Hv1h/CPP-1838, and tested the hypothesis that fusion to CPP-1838 could enhance the contact aphidicidal efficacy of HxTx-Hv1h. Additionally, we assessed the insecticidal efficacy of these recombinant peptides in the presence of a surfactant, providing insights into their practical applicability. Furthermore, a secretory recombinant expression system of HxTx-Hv1h/CPP-1838 was constructed in *P. pastoris*. Collectively, our data supports the fusion of HxTx-Hv1h with CPP-1838 as a viable strategy for enhancing insecticidal efficacy, and for offering a scalable method for the high-level expression of this potent fusion biopesticide.

## 2. Results

### 2.1. Recombinant Protein Production and Purification

We constructed synthetic genes optimized for codon usage to encode HxTx-Hv1h, HxTx-Hv1h/GNA, and HxTx-Hv1h/CPP-1838 into the pKLAC1 commercial expression vector using seamless cloning technology. This ensured the production of mature proteins devoid of additional amino acid residues (Figure 1A). Fusion constructs were designed such that the HxTx-Hv1h peptide was linked to the N-terminus of GNA or CPP-1838 by a septapeptide (alanine-glutamine-alanine-alanine-alanine-lysine-alanine), previously described as a rigid linker [36]. Each construct included a carboxy-terminal hexahistidine tag for facilitated affinity purification and glycine (G) and serine (S) residues at the N-terminus to enhance expression levels, as reported earlier. After amplification in *E. coli* and sequence confirmation, plasmids were linearized using *Sac* II and transformed into *K. lactis* competent cells. To screen for potentially enhanced protein secretion through multiple genomic insertions, PCR was performed with Integration Primers 2 and 3. SDS-PAGE analysis of culture supernatants from small-scale fermentations of these multi-copy clones led to the selection of the best performers for bench-top fermentation, enabling the generation of sufficient protein yields for subsequent bioassays. Recombinant proteins were purified from the supernatant via nickel-affinity chromatography, followed by ultrafiltration and lyophilization.

Analysis of the recombinant proteins by SDS-PAGE revealed that HxTx-Hv1h migrated as a broad band within the 10–17 kDa range, exceeding its anticipated size of 6.41 kDa (Figure 1B). Anti-His (Figure 1C) and anti-HxTx-Hv1h (Figure 1D) antibody reactivity confirmed the identity of this band as the recombinant toxin. These observations align with previous reports by Fitches et al. and Sukiran et al. on expressing HxTx-Hv1h in *P. pastoris* [12,22]. We postulate that the observed hyperglycosylation results in a mass greater than predicted. The HxTx-Hv1h/GNA fusion produced a band around 20 kDa on SDS-PAGE (Figure 1B), slightly larger than the predicted 18 kDa, and was recognized by both anti-His (Figure 1C) and anti-HxTx-Hv1h (Figure 1D) antibodies. Conversely, HxTx-Hv1h/CPP1838 migrated as a distinct band near 17 kDa, bridging the sizes of the HxTx-Hv1h and HxTx-Hv1h/GNA (Figure 1B). Western blots corroborated these findings, with immunogenic responses aligning with corresponding SDS-PAGE bands, confirming the presence of full-length HxTx-Hv1h/CPP-1838 fusion protein (Figure 1B–D). In the immunoreactivity with the anti-His antibody, an additional band was observed for both recombinant HxTx-Hv1h/GNA and HxTx-Hv1h/CPP-1838. This could be due to the presence of impure proteins that are capable of non-specific binding to anti-His antibody. Additionally, in the immunoblot with anti-HxTx-Hv1h, a lower molecular weight band was seen for HxTx-Hv1h/GNA. This suggests that the HxTx-Hv1h/GNA fusion protein may have undergone partial degradation during the expression, purification, or storage processes. In the fermentation scale-up, *K. lactis* cells generated over 5 g/L of HxTx-Hv1h in culture supernatant, as shown in Appendix A. However, fusion with GNA or CPP1838 substantially decreased expression levels to approximately 50 mg/L. Despite the differing expression yields, our results confirmed the successful recombinant production of HxTx-Hv1h and its fusions in the *K. lactis* platform. Moreover, the yeast’s low secretion of endogenous proteins facilitated the attainment of recombinant protein purity exceeding 95% after a single nickel-affinity chromatography step.

### 2.2. Contact Aphidicidal Activity of Recombinant Proteins

This study employed *M. crassicauda* as a model to evaluate the contact toxicity of recombinant proteins. Following a methodology adapted from the Chinese laboratory bioassay standards for pesticides, aphids were immersed in solutions of purified recombinant proteins at concentrations ranging from 50 µg/mL to 2000 µg/mL. We aimed to assess whether a penetration peptide could enhance the contact toxicity of HxTx-Hv1h. Previous studies have shown that GNA significantly boosts both the oral and contact toxicity of toxic peptides [12,16,22]. Consequently, we included the HxTx-Hv1h/GNA fusion as a positive control within our examinations.

As demonstrated in Figure 2, there was a dose-dependent decline in aphid survival rates following immersion in the recombinant protein solutions. Aphids immersed in buffer solutions without any recombinant protein exhibited survival rates exceeding 80%. For HxTx-Hv1h, we observed an LC_50_ of 860.5 µg/mL with a 95% confidence interval (CI) of 664.9–1007.1 µg/mL (Figure 2A). Consistent with prior studies, HxTx-Hv1h/GNA displayed notably enhanced contact toxicity, with an LC_50_ of 683.5 µg/mL (CI 379.8–932.9 µg/mL) (Figure 2B). Notably, the fusion protein HxTx-Hv1h/CPP-1838 exhibited superior contact toxicity compared to HxTx-Hv1h/GNA, with an LC_50_ of 465.2 µg/mL (CI 303.3–612.3 µg/mL) (Figure 2C). While the fusion of CPP-1838 enhanced contact toxicity akin to GNA, the contact insecticidal activity of these fusion proteins still did not reach the potential for practical applications.

Our investigations into cuticle penetration factors revealed that the hydrophobic nature of aphid cuticles impedes the effective contact of applied pesticides. Surfactants like Silwet L-77 have been shown to amplify the topical toxicity of HxTx-Hv1h against *Drosophila suzukii* by enhancing cuticle penetration [25]. After confirming the non-toxicity of 0.1% Silwet L-77 (*v*/*v*) to *M. crassicauda*, this composition was adopted as the solvent for recombinant protein solutions. The presence of Silwet L-77 led to a significant reduction, up to seven-fold, in the LC_50_ values of the tested proteins (Figure 2D–F). For example, the LC_50_ for HxTx-Hv1h/CPP-1838 drastically dropped to 66.7 µg/mL (CI 41.7–86.9 µg/mL) with the surfactant (Figure 2F). Similarly, the addition of the surfactant significantly reduced the LC_50_ values for HxTx-Hv1h and HxTx-Hv1h/GNA to 288.9 µg/mL (CI 169.1–389.5 µg/mL) and 112.3 µg/mL (CI 49.9–167.8 µg/mL), respectively (Figure 2D,E). These findings confirm that the surfactant played a critical role in enhancing the contact aphidicidal activity of the recombinant proteins. Overall, the fusion with CPP1838 consistently enhanced the toxicity of HxTx-Hv1h, and inculcation of the surfactant markedly improved the contact aphidicidal activity across all the proteins tested.

### 2.3. Detection of Recombinant Proteins within Aphid Bodies

To evaluate the potential of recombinant proteins to penetrate aphid cuticles, *M. crassicauda* were exposed to contact treatments with each recombinant protein at a concentration of 200 µM. Following an 8 h incubation period, the aphids underwent surface cleaning procedures prior to the extraction of total proteins, which were analyzed using Western blotting. Due to specificity issues with the anti-His antibody in whole aphid protein extracts, the detection was performed using the anti-HxTx-Hv1h antibody. The analysis, presented in Figure 3, revealed positive immunoreactivity in extracts from aphids treated with HxTx-Hv1h, HxTx-Hv1h/GNA, and HxTx-Hv1h/CPP-1838, demonstrating that these recombinant proteins were able to penetrate the aphid cuticle irrespective of the presence of a surfactant. Notably, treatments involving HxTx-Hv1h/GNA and HxTx-Hv1h/CPP-1838 exhibited stronger immunoreactivity compared to those with HxTx-Hv1h alone, regardless of surfactant inclusion. This difference is consistent with previous findings where GNA was shown to enhance the internalization and persistence of HxTx-Hv1h following topical application [12,22]. The results suggest that CPP-1838 may similarly enhance the toxicity of HxTx-Hv1h, following a mechanism comparable to that of GNA.

With regard to the influence of surfactants, incorporating Silwet L-77 significantly increased the absorption of HxTx-Hv1h by the aphids. Although the changes were not as pronounced for HxTx-Hv1h/GNA and HxTx-Hv1h/CPP-1838, there was a notable trend toward improved stability within the aphid bodies. In experiments without surfactant, aphid protein extracts from the groups treated with HxTx-Hv1h/GNA and HxTx-Hv1h/CPP-1838 showed more pronounced bands of immunoreactivity. Conversely, after adding surfactant, fewer immunoreactive bands were observed (Figure 3). Overall, the addition of surfactant not only enhanced the efficiency of HxTx-Hv1h delivery to the target site but also suggested that stability plays a crucial role in the enhanced toxicity observed for HxTx-Hv1h/GNA and HxTx-Hv1h/CPP-1838. This study highlights the relevance of both delivery efficiency and protein stability in enhancing the contact toxicity of recombinant proteins against aphids.

### 2.4. Construction of HxTx-Hv1h/CPP-1838 Multicopy Expression Cassettes

To enhance the expression levels of the recombinant protein HxTx-Hv1h/CPP-1838, which exhibited potent aphidicidal activity when applied with surfactants, we explored the generation of *P. pastoris* clones harboring varying copy numbers of the gene of interest. Increasing recombinant protein yield can frequently be achieved by raising the gene copy number within the host cells. We aimed to ascertain the gene dosage that would result in maximal protein expression.

The initial step involved seamlessly cloning the HxTx-Hv1h/CPP-1838 gene sequence downstream of the secretion signal peptide within the pGAPZαA vector to produce the construct pGAPZαA-HxTx-Hv1h/CPP-1838-1C, which carried a single copy of the expression cassette (Appendix A). We devised a method for creating vectors with multiple gene copies, as shown in Appendix A. The pGAPZαA-HxTx-Hv1h/CPP-1838-1C vector was subjected to double digestion using two sets of restriction enzymes, *Mlu* I/*Bgl* II and *Mlu* I/*Bam*H I. The backbone fragment of the construct, which lacked the pUC origin of replication, was retrieved through gel extraction after digestion with *Mlu* I/*Bgl* II. Concurrently, the fragment comprising the expression cassette and pUC origin was isolated via *Mlu* I/*Bam*H I digestion. By ligating these two fragments, we obtained the pGAPZαA-HxTx-Hv1h/CPP-1838-2C construct, which contained two copies of the expression cassette.

Following a similar workflow, constructs harboring 4, 6, 8, 10, and 12 copies of the expression cassette were synthesized and designated as pGAPZαA-HxTx-Hv1h/CPP-1838-4C, -6C, -8C, -10C, and -12C, respectively. Analysis using 1% agarose gel electrophoresis confirmed that the size of the recombinant constructs escalated proportionally with the number of expression cassettes incorporated (Figure 4A). Verification of the constructs with different copy numbers was further achieved through *Bgl* II/*Bam*H I double enzyme digestion. This produced two distinct bands for each construct; one of the bands remained constant in size, while the other increased incrementally with each additional expression cassette included in the constructs (Figure 4B). These outcomes substantiated the successful fabrication of expression vectors with multitude gene copies for HxTx-Hv1h/CPP-1838.

### 2.5. Transformation and Gene Copy Analysis in Recombinant P. pastoris

To establish recombinant *P. pastoris* strains capable of expressing the HxTx-Hv1h/CPP-1838 protein, we transformed the linearized multi-copy constructs, prepared using *Bgl* II restriction enzyme, into the *P. pastoris* X-33 strain through electroporation. Transformants were selected on YPD agar plates containing 200 mg/L zeocin and incubated at 30 °C for three days. The resultant recombinant *P. pastoris* clones were designated according to the number of targeted gene copies they potentially harbored: 1C, 2C, 4C, 6C, 8C, 10C, and 12C. To accurately determine the copy number of the HxTx-Hv1h/CPP-1838 gene in the yeast genome, qPCR was employed, comparing the target gene to an endogenous gene present in a single copy within the genome. The analysis, displayed in Figure 4C, encompassed six confirmed transformants for each set of clones. The qPCR data revealed that under the applied selection pressure conditions, clones purported to be single-copy typically carried only one integrated gene of the recombinant protein. However, within the alleged multi-copy clones, not every transformant contained the corresponding number of gene copies denoted by their initial construct; for instance, among the 8C cohort, merely half of the transformants comprised eight copies of the gene. The rest embodied lower copy numbers—two, four, or six—suggesting variability in integration events. These observations highlight the importance of rigorous screening post-transformation, particularly when using multi-copy expression vectors. Although achieving complete integration of multi-copy constructs is plausible, losses in copy numbers during the transformation process can occur. Hence, the successful adoption of a multi-copy approach to maximize recombinant protein production in *P. pastoris* relies on effectively identifying and selecting clones with the desired number of gene integrations.

### 2.6. HxTx-Hv1h/CPP-1838 Protein Expression from P. pastoris Clones

The recipe for many refinements of recombinant protein expression levels typically hinges on the integration of the gene copy number in yeast transformants, yet expression often varies among individual clones. In our investigation, we initially screened multiple *P. pastoris* clones for the expression of HxTx-Hv1h/CPP-1838 under standardized small-scale conditions and identified the top two high-expressing transformants at each copy number (unpresented results). We then sought to elucidate any existing correlation between the gene copy number and the expression level of HxTx-Hv1h/CPP-1838. Following three days of culture, SDS-PAGE analysis of the culture supernatants from the 1C through 12C transformants revealed a discernible trend: the expression level of HxTx-Hv1h/CPP-1838 rose concomitantly with increases in the gene copy number (Figure 5A). Furthermore, the highest expression levels were observed in clones containing ten copies of the gene. Beyond this threshold, a paradoxical decrease in protein yield was observed, suggesting that blindly augmenting the gene copy number is not a guarantee for amplified expression.

Additionally, the clone that demonstrated the acme of expression during the screening underwent bench-top fermentation. Analyses via SDS-PAGE at eight-hour intervals indicated the dominance of the HxTx-Hv1h/CPP-1838 recombinant protein in the fermentation supernatant—it markedly comprised over 50% of the total protein (Figure 5B). A subsequent assessment of protein concentrations using the BCA method, corroborated by standard protein concentration comparison, surmised that the HxTx-Hv1h/CPP-1838 yield after 56 h of fermentation approached an impressive 2.30 g/L (Figure 5B). In summary, these results corroborate that a strategic increase in the gene copy number to an optimal level of ten, coupled with meticulous fed-batch fermentation, can culminate in the robust expression of HxTx-Hv1h/CPP-1838.

## 3. Discussion

The escalating environmental concerns linked to the use of synthetic chemical pesticides have led to a pivotal shift towards biological controls, positioning them as a greener alternative insect management strategy. In the evolving field of venomics, it has been discovered that the venom of predatory insects harbors a wealth of neurotoxic peptides, positioning these venoms as a veritable goldmine for the development of protein-based biopesticides [10,11,37]. However, for a neurotoxic peptide to be applicable as a biopesticide, it must meet several prerequisites: first, it must possess oral or contact insecticidal activity; second, stability, being able to tolerate a wide range of temperatures and pH levels; third, cost-effectiveness, which includes efficient expression in a recombinant expression host; and fourth, selectivity, meaning that the toxic peptide should be safe for humans and beneficial to the natural enemies of pests. It is because of these numerous barriers that the exploration for neurotoxic peptides suitable for biopesticide development remains a hot topic in the study of protein-based biopesticides. HxTx-Hv1h meets the aforementioned criteria and is the first neurotoxic peptide derived from spider venom to be developed into a commercial biopesticide to date. However, numerous studies have shown that despite the oral or contact activity of HxTx-Hv1h, its potency requires further enhancement. Previously, lectins, particularly GNA, have primarily been utilized as a “carrier” for the neurotoxic peptide HxTx-Hv1h to improve the efficiency of transport to its site of action, thereby increasing its oral or contact insecticidal activity. CPPs are typically employed in the delivery of molecular cargos into human cells, and have only recently been considered in the context of enhancing insecticides [31,38]. This study introduces yeast as the production host for a fusion protein that combines HxTx-Hv1h with the N-terminus of CPP-1838, subsequently investigating its enhanced aphid-killing effects, with and without surfactant assistance, to affirm the role of CPP-1838 as a promising biopesticide booster. Our research demonstrates that CPP-1838 can match or even surpass GNA in enhancing the contact aphidicidal effect of HxTx-Hv1h, marking a novel discovery that CPP can significantly bolster the potency of protein-based biopesticides. The successful construction and hefty expression of multi-copy HxTx-Hv1h/CPP-1838 in *P. pastoris* pave the way for its larger-scale utilization for biopesticide applications.

Its robust secretory expression mechanisms and capability for post-translational modifications render yeast an exemplary host for expressing exogenous proteins, particularly those with complex disulfide bonds like HxTx-Hv1h. In our approach, we opted for *K. lactis* and *P. pastoris* as our expression hosts. These yeasts facilitate the attainment of pure recombinant proteins following a single affinity chromatography purification step. Whilst the actual size of the recombinant proteins deviated from the theoretical molecular weight, the recombinant proteins had immunoreactivity with anti-His and anti-HxTx-Hv1h antibodies, and Western blot analysis results corresponded with SDS-PAGE, providing evidence that these three proteins were successfully expressed in yeast. Of course, whether the recombinant expression of HxTx-Hv1h in yeast was completely correct, such as whether disulfide connectivity was preserved, still needs further elucidation through studies comparing the toxicity of recombinant HxTx-Hv1h with natural toxin. The study found that regardless of which yeast chassis was chosen, the recombinant expression level of HxTx-Hv1h was relatively high, but the recombinant expression level dropped sharply when HxTx-Hv1h was co-expressed with GNA or CPP-1838, suggesting that the translation efficiency of HxTx-Hv1h was greatly reduced when expressed as a fusion with other peptide segments.

The neurotoxic peptide HxTx-Hv1h, which is drawn from spider venom, exhibits both oral and contact toxicity to a range of aphid species. For the purpose of this study, *M. crassicauda* was utilized as the representative test insect, with bioactivity assays carried out through immersion methods. A solution containing 1000 μg/mL of the recombinant HxTx-Hv1h induced greater than 60% mortality within two days, with the calculated LC_50_ against *M. crassicauda* pegged at 860.5 μg/mL. The bioactive peptide demonstrated the capability to transverse the aphid cuticle, as evidenced by Western blot analysis detecting its presence at intended target sites within the insect. This result complements the findings of Sukiran et al., who reported LC_50_ values of HxTx-Hv1h against pea and peach-potato aphids at 0.70 and 0.68 mg/mL, respectively [12]. Noteworthy is the variance in efficacies, presumably attributable to variations in the pathways and mechanisms of delivery, as well as the diverse receptor affinities observed across different aphid species. Furthermore, mannose-binding lectins, particularly GNA, when employed as peptide transport carriers, have been shown to bolster the oral and contact effects of insecticidal peptides. The production of the fusion protein HxTx-Hv1h/GNA allowed us to assess this synergy in the current study. Through bioassay and immunoblot investigations, we discerned that GNA facilitates the cuticle penetration of HxTx-Hv1h, thereby enhancing the peptide’s aphidicidal action—a phenomenon endorsed by prior studies [12,13,16,22,23]. Additionally, the literature suggests that GNA could contribute to the prolonged stability and efficacy of HxTx-Hv1h post-ingestion [12,13]. In a comparative light, CPP-1838 demonstrated superior amplification of the HxTx-Hv1h contact toxicity, evidencing stronger immunoreactivity sans additional surfactants in anti-HxTx-Hv1h antibody assays. Such augmentation is presumably a result of CPP-1838’s amphiphilic nature, promoting easier penetration through the hydrophobic insect cuticle layers. However, it remains to be clarified whether the increased uptick in delivery efficiency can be singularly ascribed to CPP-1838, as its potential intrinsic toxicity towards insects requires further substantiation.

Although fusing HxTx-Hv1h with GNA or CPP-1838 significantly enhanced its aphidicidal activity, it is still far from use in practical applications. Adjuvants, capable of shuttling active compounds into the insect hemolymph via paracellular routes, are critical in accentuating the bio-efficacy of such active elements [25,39]. Nonetheless, the efficacy of adjuvants is not uniform across active ingredients, exhibiting considerable variability. A notable study by Fanning et al. elucidated the symbiotic interplay between HxTx-Hv1h and surfactants like Silwet L-77 in targeting spotted-wing *Drosophila*, highlighting Silwet L-77’s minimal insecticidal footprint but pronounced competence in cuticle penetration—attributes that synergize well with HxTx-Hv1h [25]. Capitalizing on this insight, our study incorporated a dose of Silwet L-77, benign to aphids, into our recombinant protein solutions, charting a significant elevation—threefold, sixfold, and sevenfold—in aphidicidal activity across HxTx-Hv1h, HxTx-Hv1h/GNA, and HxTx-Hv1h/CPP-1838, respectively. Western blot findings suggest that this augmentative effect could be twofold: enhancing delivery while concurrently offering a protective buffer against insect proteolytic enzymes, thereby reducing recombinant protein degradation. In summation, Silwet L-77 harbors the potential to boost the contact effectiveness of protein-based bioinsecticides targeting aphids.

With the observed low recombinant expression level of HxTx-Hv1h/CPP-1838, this study aimed to enhance expression within the yeast *P. pastoris* host system. Methods to elevate recombinant protein production are multifaceted, encompassing codon optimization, gene copy number augmentation, judicious selection of the host strain and promoter, signal sequence engineering, and overall process refinement. This investigation specifically delved into the impact of gene dosage on yielding elevated levels of HxTx-Hv1h/CPP-1838 in the *P. pastoris* system. Typically, to boost the gene dosage, researchers may either increase antibiotic concentrations to select high-copy genetic transformants [40] or link multiple exogenous gene cassettes via isocaudamers to construct a multi-copy vector for genomic integration [33,34,35]. However, the former approach proved to be a less efficient, untargeted strategy. The current study improved upon the established *Bgl*Brick technique, developing in vitro multimers with up to twelve expression cassettes using an additional restriction endonuclease, *Mlu* I, alongside *Bgl* II and *Bam*H I. This sophisticated assembly revealed that an increased construct size corresponded to a reduced yield of *E. coli* transformants and decelerated growth—presumably a consequence of hindered resistance gene expression with larger vectors. Remarkably, such limitations were not observed in *P. pastoris*, attesting to its favorable eukaryotic machinery for handling substantial multi-copy vectors. Post-transformation of *P. pastoris* with varied copy number cassettes, the precise integration dosage in the clone genomes was determined utilizing qPCR. Findings indicated incomplete multi-copy integration in certain cases, corroborating earlier insights by Pyati et al. [34]. Subsequent shake flask cultures demonstrated a positive correlation between escalating gene dosage and HxTx-Hv1h/CPP-1838 production, peaking at a ten-copy construct. Notably, any further amplification of gene copies inversely impacted expression levels—a finding aligned with the experiences documented by Dagar et al. in interleukin-3 production [33]. Similarly, Pyati et al. suggested that increasing the gene copy number to nine was optimal for recombinant expression of Hv1a/GNA, and further increasing the copy number would decrease its expression level [34]. Benchtop fermenter culturing of the highest-yielding transformant under high cell-density, fed-batch fermentation conditions culminated in a yield of 2.3 g/L of HxTx-Hv1h/CPP-1838 after 56 h. We posit that the recombinant expression of HxTx-Hv1h/CPP-1838 can be further enhanced through medium optimization and refinement of key fermentation parameters and processes, setting a promising stage for future application scalability.

## 4. Conclusions

In conclusion, this study presents the first evidence that fusion of the neurotoxic peptide HxTx-Hv1h with the cell-penetrating peptide CPP-1838 significantly improves its efficacy against aphids. The synergistic effect with surfactant Silwet L-77 facilitates enhanced cuticular penetration, leading to superior contact aphidicidal performance. Our findings with the *Bgl*Brick methodology also indicate that increased gene copy numbers in *P. pastoris* cells of up to ten copies correlate with higher expression levels of HxTx-Hv1h/CPP-1838. High-density fermentation successfully yielded 2.30 g/L of the active fusion protein, underscoring the potential of CPPs to enhance the effectiveness of protein-based biopesticides.

## 5. Materials and Methods

### 5.1. Materials

*Escherichia coli* DH5α competent cells were sourced from Sangon Biotech (Shanghai, China). *K. lactis* strain GG799 and the expression vector pKLAC1 were acquired from New England Biolabs (NEB) (Ipswich, MA, USA). *P. pastoris* expression strain X-33 and the plasmid pGAPZαA, along with antibiotic zeocin, were purchased from Invitrogen (Carlsbad, CA, USA). Primers and gene sequences encoding HxTx-Hv1h, HxTx-Hv1h/GNA, and HxTx-Hv1h/CPP-1838 was synthesized by General Biol Ltd. (Chuzhou, China). Restriction enzymes, T4 DNA ligase, various DNA polymerases, SYBR Premix Ex Taq kit, and In-fusion^®^ snap assembly master mix were procured by Takara (Kyoto, Japan). Polyclonal anti-HxTx-Hv1h and monoclonal anti-His-tag antibodies were prepared by Gene Create Ltd. (Wuhan, China). Horseradish peroxidase-conjugated secondary IgG antibodies were obtained from Bio-Rad (Hercules, CA, USA). Plasmid isolation kits, gel extraction kits, yeast gDNA extraction kit, and BCA protein assay kits were provided by Sangon Biotech. The yeast carbon base (YCB) medium required for culturing yeast was furnished by NEB. Components of fermentation medium and other molecular biology-grade reagents were all purchased from local suppliers.

### 5.2. Construction of Recombinant Expression Vectors

The cloning of the coding sequences for the neurotoxic peptides minus any additional amino acids was achieved through strategic construction on expression vectors pKLAC1 and pGAPZαA, employing a seamless cloning technique [41]. Positive transformants selected by antibiotic screening were verified by colony polymerase chain reaction (PCR) and then further verified by DNA sequencing. Yeast codon-optimized nucleotide sequences and amplification primers are shown in Appendix A.

### 5.3. Assembly of Multicopy Expression Cassettes

Details of the pGAPZαA vector containing a single copy of the HxTx-Hv1h/CPP-1838 expression cassette are shown in Appendix A. The HxTx-Hv1h/CPP-1838 expression cassette was neatly inserted between the *Bam*H I and *Bgl* II sites in the vector backbone (Appendix A). Utilizing the *Bgl*Brick assembly method [33], we constructed multiple expression cassettes carrying varying gene copy numbers by exploiting the isocaudomer relationship between the *BamH* I and *Bgl* II restriction sites, creating a hybrid site that is resistant to cleavage by either enzymes. The single-copy pGAPZαA-HxTx-Hv1h/CPP-1838-1C was subjected to double-digestion using *Mlu* I/*Bam*H I to release the expression cassette and pUC ori, while *Mlu* I/*Bgl* II digestion was used to retrieve the vector backbone. Overnight ligation at 16 °C was conducted with the purified expression cassette and vector backbone, generating recombinant constructs containing an incrementally increased copy number of the expression cassettes (2C, 4C, 6C, 8C, 10C, and 12C), where “C” indicates the number of copies. After each ligation, the mixture was transformed into *E. coli* DH5α competent cells, and zeocin-resistant clones were identified on lysogeny broth (LB) agar plates containing 30 mg/L zeocin for primary screening. Positive clones were confirmed firstly by colony PCR. Further validations included double-digestion analysis and DNA sequencing to ensure the constructs harbored the desired number of cassette copies in the correct orientation and sequence fidelity. Appendix A portrays the methodical steps undertaken for the assembly of multi-copy expression cassettes within the pGAPZαA vector backbone, corresponding to each increase in gene copy number.

### 5.4. Yeast Transformation and Screening

The integration and confirmation of the HxTx-Hv1h-derived recombinant constructs in *K. lactis* and *P. pastoris* were performed following specific molecular biology protocols, as described below.

Integration into *K. lactis*: Plasmids pKLAC1-HxTx-Hv1h, pKLAC1-HxTx-Hv1h/GNA, and pKLAC1-HxTx-Hv1h/CPP-1838 were each digested with *Sac* II to verify proper insert integration through electrophoretic analysis. Complete digestion was assessed via 1% agarose gel electrophoresis. The purified linearized fragments were then transformed into the *K. lactis* strain GG799 competent cells via electroporation using a Micropulser Electroporator, following the established methods of Wolf et al. [42]. Successful transformants were selected on YCB agar medium supplemented with 5 mM acetamide. Integration of the expression cassettes into the genome was confirmed by colony PCR with the appropriate integration primers (Primers 1 and 2 mentioned), while whole-cell PCR using Primers 2 and 3 helped identify multi-copy integrants, as detailed in Appendix A.

Integration into *P. pastoris*: pGAPZαA-HxTx-Hv1h/CPP-1838-1C and other multi-copy constructs were linearized using *Bgl*II. Following gel electrophoresis and DNA extraction, the linearized constructs were transformed into *P. pastoris* X-33 competent cells utilizing protocols described previously [43]. Transformed cells were plated on yeast extract peptone dextrose (YPD) agar containing 200 mg/L zeocin for selection. Colony PCR with pGAP forward and 3′AOX1 primers, detailed in Appendix A, was used to identify positive transformants. Finally, quantitative PCR (qPCR) was performed to quantify the integration of the HxTx-Hv1h/CPP-1838 cassettes into the *P. pastoris* genome, confirming the presence of single or multiple copies.

### 5.5. Copy Number Analysis of P. pastoris Transformants by qPCR

Transformants of *P. pastoris* harboring the gene of interest were cultured in 5 mL YPD medium supplemented with 200 mg/L zeocin and incubated at 30 °C with agitation until an OD_600_ of 8–10 was achieved. Genomic DNA was extracted from these cultures using a commercial extraction kit, according to the manufacturer’s instructions. qPCR assays were performed in triplicate for each sample using a 20 μL reaction volume. Each reaction mixture contained 10 μL of 2× SYBR^®^ Premix Ex Taq Mix, 0.5 μM of each forward and reverse primer, and 10 ng of template DNA. Primers were designed to specifically amplify the fragments of the HxTx-Hv1h/CPP-1838 gene and the endogenous β-actin gene of *P. pastoris*, which served as a reference control for gene copy number normalization (primer sequences are displayed in Appendix A) [44]. The thermal cycling conditions were programmed into the qTower 3G Real-Time PCR System (Analytik Jena, Germany) and comprised an initial denaturation step at 95 °C for 5 min, followed by 40 amplification cycles consisting of 20 s at 95 °C, 20 s at 60 °C, and 20 s at 72 °C. To construct the standard curve required for gene copy number estimation, genomic DNA from the wild-type *P. pastoris* strain and the recombinant pGAPZαA-HxTx-Hv1h/CPP-1838-1C construct, each containing a known single copy of the target gene, were employed. The cycle threshold (Ct) values obtained from the qPCR assay were plotted against the standard curve to determine the relative gene copy number in the transformant samples.

### 5.6. Screening Transformants Expressing Recombinant Protein

Selected clones of *K. lactis* transformants were cultured in YPGal medium (10 g/L yeast extract, 20 g/L peptone, 20 g/L galactose) at 30 °C with shaking, for 72 h on a small scale (5 mL). Cultures were then centrifuged at 12,000× *g* for 2 min to pellet the cells. From the supernatant, 30 μL was sampled and resolved on a 15% SDS-PAGE gel to analyze protein expression. Protein bands were visualized following electrophoresis using Coomassie G-250 ultrafast staining solution (comprising 6% ammonium sulfate, 1.5% β-cyclodextrin, 15% ethanol, 0.015% Coomassie G-250, with the pH adjusted to 1.3). Transformants of *P. pastoris* expressing HxTx-Hv1h/CPP-1838 were grown in YPD medium containing 200 mg/L zeocin. After a 3-day incubation at 30 °C, the cultures were centrifuged, and the resulting supernatant was subjected to SDS-PAGE analysis to identify those lines with robust expression of HxTx-Hv1h/CPP-1838.

### 5.7. Bench-Top Fermentation

Transformants of both *K. lactis* and *P. pastoris* were cultivated in bench-top bioreactors utilizing a 10 L capacity Intelli-FermB (T&J Bioengineering Co., Ltd., Shanghai, China). While the general fermentation strategy remained identical for both species, two significant differences were noted. Firstly, *K. lactis* was cultured in YCB medium supplemented with 5 mM acetamide as the seed medium, whereas *P. pastoris* utilized YPD. Secondly, the supplemental carbon sources differed: galactose was employed for *K. lactis* while glucose was used for *P. pastoris*. The fermentation medium was prepared based on methodologies detailed in previous reports [39]. Both fermentations were maintained under the following identical conditions: the pH was kept constant at 5.0 using 28% ammonium hydroxide, and the culture temperature was regulated at 28 °C. A minimum of 40% dissolved oxygen (DO) was maintained, managed via cascade agitation and airflow adjustments. A concentrated feed containing 1000 g/L of the appropriate carbon source (galactose for *K. lactis* and glucose for *P. pastoris*) along with 20 mL/L PTM1 salts was introduced at a flow rate of 50 mL/h. Samples were periodically collected at various fermentation stages. Growth metrics were evaluated by measuring the wet weight of the pelleted cells. The expression of the recombinant protein was assessed qualitatively and quantitatively at different time points during the fermentation process.

### 5.8. Purification, Quantification, and Western Blot Analysis of Recombinant Proteins

The recombinant proteins HxTx-Hv1h, HxTx-Hv1h/GNA, and HxTx-Hv1h/CPP-1838 were purified directly from the yeast fermentation supernatant. Using His-tag affinity chromatography, purification was achieved in a single step utilizing 30 mL Ni-NTA gravity columns (Smart-Lifesciences, Changzhou, China). The columns were first equilibrated with binding buffer (25 mM Tris, 300 mM NaCl, pH 8.0). The supernatant was then applied to the column, followed by the application of 2 to 5 column volumes of wash buffer (25 mM Tris, 300 mM NaCl, 30 mM Imidazole, pH 8.0) to eliminate non-specifically bound proteins. Elution was performed using an elution buffer (25 mM Tris, 300 mM NaCl, 200 mM Imidazole, pH 8.0), and the eluates containing the target proteins were concentrated via ultrafiltration and subsequently lyophilized for further analyses. The concentration of the purified proteins was determined via SDS-PAGE, supported by a BCA protein assay with bovine serum albumin (BSA) as the reference standard. Protein samples were appropriately diluted to ensure the accuracy of spectrophotometric readings within the linear range of the assay. The resolved proteins on 15% acrylamide SDS-PAGE gels were transferred onto nitrocellulose membranes using a semi-dry transfer cell (Trans-blot^®^ SD, Bio-Rad, USA), following standard protocols. Immunodetection was performed utilizing anti-His-tag antibodies at a 1:5000 dilution and anti-HxTx-Hv1h antibodies at a 1:1000 dilution to verify the presence and integrity of the recombinant proteins.

### 5.9. Aphid Rearing and Contact Toxicity Bioassays

*Megoura crassicauda* aphids were cultured on broad bean plants under controlled conditions at 22 °C, with a 16 h light/8 h dark photoperiod. The bioassay was conducted according to the methods outlined in the Chinese pesticides guidelines for laboratory bioactivity tests, Part 6, which focuses on the immersion test for insecticide activity, albeit with minor adaptations. Broad bean seedlings, approximately 4 cm in height, were inoculated with aphids. Following colonization and propagation over three generations (approximately 3 days post-inoculation), the seedlings’ roots were trimmed, and excess aphids were carefully removed using a drawing pen, retaining 20–30 wingless adult aphids of uniform size per seedling. The prepared seedlings were then immersed for 10 s in solutions containing various concentrations of recombinant proteins, diluted in sterile water or 0.1% Silwet L-77—an adjuvant whose primary component is 99.5% polyalkyleneoxide-modified heptamethyltrisiloxane. Following immersion, excess solution was blotted from the seedlings using filter paper. To maintain moisture, the treated seedlings were embedded in 1% agar and returned to the original culturing conditions. Both sterile water and 0.1% Silwet L-77 served as controls. Three biological replicates were conducted for each treatment concentration. Aphid mortality was assessed 48 h after treatment. Both the total number of aphids and the number of deceased aphids were recorded.

### 5.10. Statistical Analysis

Data management and statistical analyses were carried out using Excel software (Microsoft, Redmond, CA, USA) and GraphPad Prism version 8 (San Diego, CA, USA). Differences between individual data points were assessed using Student’s *t*-test to determine statistical significance. The LC_50_ values from bioassay experiments were calculated through logistic regression of the log-transformed concentration–response data.

## Figures and Tables

**Figure 1 toxins-16-00358-f001:**
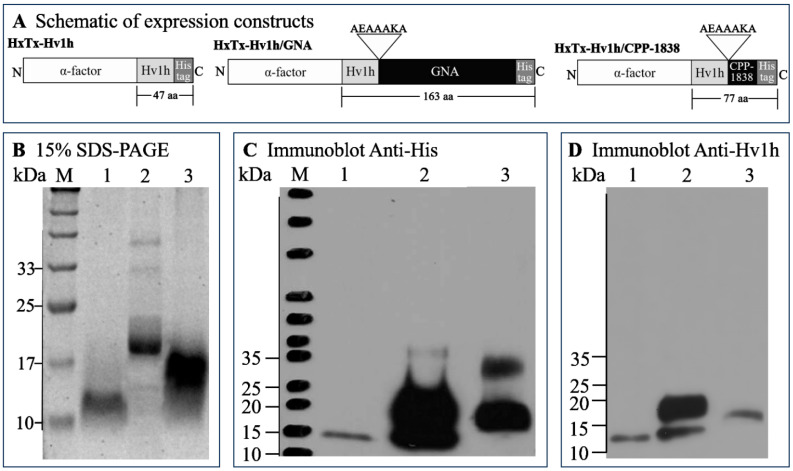
Expression and detection of recombinant proteins. (**A**) Schematic representation of constructs engineered to encode HxTx-Hv1h, HxTx-Hv1h/GNA, and HxTx-Hv1h/CPP-1838 in yeast. The α-factor leader sequence facilitates targeting of the expressed proteins to the yeast secretory pathway, thus enabling their isolation from the fermentation culture supernatant. The His-tag indicates an incorporated hexahistidine motif, which facilitates protein purification via nickel-nitrilotriacetic acid (Ni-NTA) affinity chromatography and permits detection through Western blot analysis. (**B**) Resolution of purified recombinant proteins on a 15% SDS-PAGE gel, visualized post-staining with Coomassie ultrafast. ‘M’ designates the molecular weight marker; lane 1 contains 5 μg of HxTx-Hv1h; lane 2 contains 7.5 μg of HxTx-Hv1h/GNA; lane 3 features 10 μg of HxTx-Hv1h/CPP-1838. (**C**) Immunoblot analysis of the recombinant proteins utilizing an anti-His tag antibody. ‘M’ comprises molecular weight standards, with lanes 1–3 corresponding to the samples detailed in (**B**), loaded with approximately 50 ng of HxTx-Hv1h, 200 ng of HxTx-Hv1h/GNA, and 100 ng of HxTx-Hv1h/CPP-1838, respectively. (**D**) Western blot detection of the recombinant proteins using an anti-HxTx-Hv1h antibody. Lanes 1–3 follow the pattern established in (**B**), with loading quantities at approximately 50 ng for HxTx-Hv1h and HxTx-Hv1h/CPP-1838, while 100 ng of HxTx-Hv1h/GNA is introduced in lane 2.

**Figure 2 toxins-16-00358-f002:**
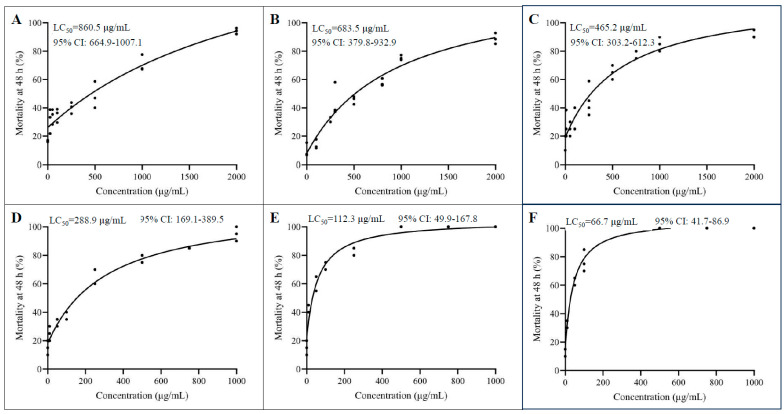
Evaluation of the contact toxicity of recombinant proteins against aphids. (**A**–**C**) Dose–response curves depicting the contact toxicity of HxTx-Hv1h, HxTx-Hv1h/GNA, and HxTx-Hv1h/CPP-1838 on aphid populations, in the absence of the surfactant Silwet L-77. The curves illustrate the percentage of mortality across a spectrum of concentrations, highlighting the aphidicidal potency of the recombinant proteins without surfactant assistance. (**D**–**F**). Dose–response curves showcasing the augmented contact toxicity of HxTx-Hv1h, HxTx-Hv1h/GNA, and HxTx-Hv1h/CPP-1838 when used in conjunction with the surfactant Silwet L-77. These curves demonstrate the synergy between the bioactive proteins and the surfactant, culminating in enhanced lethality towards the aphids.

**Figure 3 toxins-16-00358-f003:**
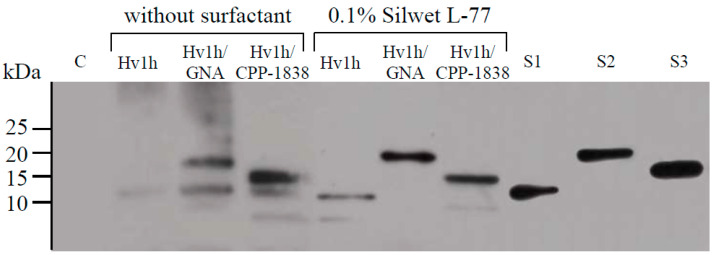
Immunoblot analysis of protein extracts from insects. ‘C’ denotes the lanes loaded with extracts from control aphids that did not undergo exposure to any recombinant proteins. Above each lane, the identity of the specific recombinant protein that the aphids contacted is labeled. The protein samples were derived from groups of 20 aphids harvested 8 h following contact with a 200 μM concentration of each respective recombinant protein. ‘S1’, ‘S2’, and ‘S3’ correspond to the protein standards for HxTx-Hv1h, HxTx-Hv1h/GNA, and HxTx-Hv1h/CPP-1838, each at a concentration of 100 ng.

**Figure 4 toxins-16-00358-f004:**
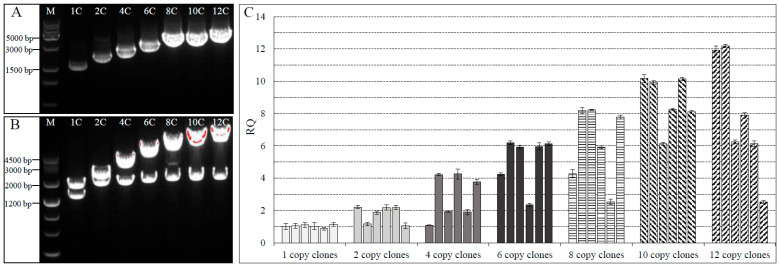
Confirmation of plasmids harboring single and multiple HxTx-Hv1h/CPP-1838 expression cassettes and copy number assessment in transformed yeast clones. (**A**) plasmid shift analysis. ‘M’ denotes the molecular weight standards. Lanes 1C through 12C contain plasmids pGAPZαA-HxTx-Hv1h/CPP-1838 with varying copy counts, notationally indicated as 1C (one copy), 2C (two copies), 4C (four copies), 6C (six copies), 8C (eight copies), 10C (ten copies), and 12C (twelve copies), respectively. (**B**) Analysis of plasmid conformation subsequent to double restriction digestion with *Bgl* II and *Bam*H I enzymes, with each lane corresponding to the respective plasmid construct detailed in (**A**). (**C**) qPCR evaluation of yeast clones transformed with plasmids encompassing single and assorted copy numbers of the HxTx-Hv1h/CPP-1838 expression cassette. The RQ (Relative Quantification) values signify the comparative determination of copy numbers of the HxTx-Hv1h/CPP-1838 construct in relation to a reference housekeeping gene.

**Figure 5 toxins-16-00358-f005:**
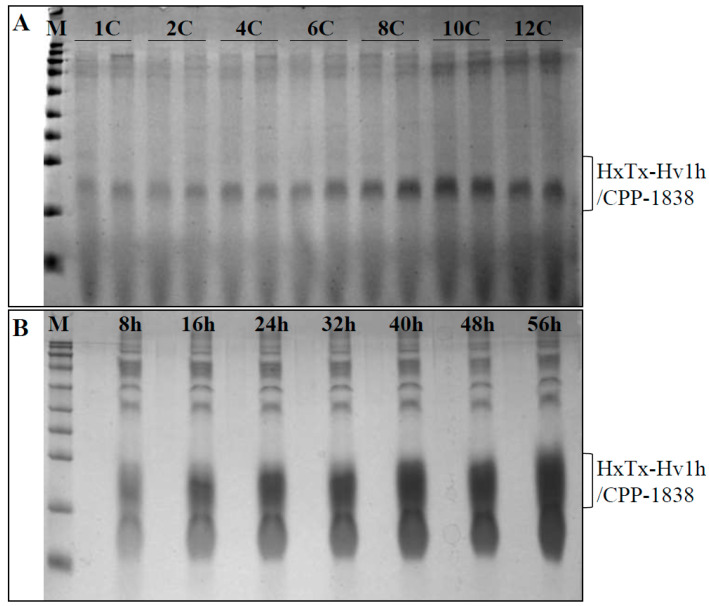
Evaluation of protein expression levels in *P. pastoris* transformants with multiple gene copies. (**A**) Depiction of cultured supernatant protein profiles from *P. pastoris* transformants incorporating varying copy numbers of the HxTx-Hv1h/CPP-1838 expression cassettes. ‘C’ represents the expression cassette featuring a GAP promoter, the HxTx-Hv1h/CPP-1838 gene, and an AOX1 terminator. ‘1C’ denotes the yeast transformant harboring a single expression cassette. ‘2C’, ‘4C’, ‘6C’, ‘8C’, ‘10C’, and ‘12C’ indicate transformants encompassing two, four, six, eight, ten, and twelve repeats of the expression cassette, respectively. (**B**) Chronological analysis of HxTx-Hv1h/CPP-1838 protein expression in yeast transformants across diverse fermentation time frames. ‘M’ represents the molecular weight marker.

## Data Availability

The raw data supporting the conclusions of this article will be made available by the authors on request.

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
