# Peer review of "Cell Penetrating Peptide Enhances the Aphidicidal Activity of Spider Venom-Derived Neurotoxin"

_toxins, 2024, doi:10.3390/toxins16080358_

Round 1
Reviewer 1 Report
Comments and Suggestions for Authors
The manuscript presents a novel approach to enhancing the aphidicidal activity of a spider venom-derived neurotoxin through fusion with cell-penetrating peptides (CPPs). The study's findings are significant, showing potential for industrial-scale recombinant synthesis of HxTx-Hv1h/CPP-1838 and its application as a biopesticide. However, several areas require attention to improve the quality of the manuscript.
The abstract provides a concise summary but could benefit from clearer statements regarding the significance of the findings. Consider highlighting the novelty of using CPPs in the context of biopesticides in the opening sentence to immediately capture the reader's interest.
The introduction is informative but dense. Dividing long paragraphs into shorter ones with clear subheadings (e.g., "Impact of Herbivorous Insects on Agriculture", "Challenges for Chemical Pesticides", "Potential for Biological Pesticides", etc.) may improve readability.
The rationale for selecting HxTx-Hv1h vs CPP-1838 should be highlighted more clearly. Why was CPP-1838 chosen?
The methods section is detailed, but some descriptions of procedures could have been shortened or moved to supplemental materials to focus on the main results. Ask about any new or particularly difficult procedures, as this will help other researchers replicate your work. The results are clearly shown in appropriate figures, but complex sentences need to be simplified for better understanding, e.g., “Vector pGAPZαA containing a single copy of the HxTx-Hv1h/CPP-1838 expression cassette is shown in detail in the supplementary figures. S1A” can be abbreviated to “Details of the pGAPZαA vector containing a single copy of the HxTx-Hv1h/CPP-1838 expression cassette are shown in Supplementary Figure S1A.”
The discussion should more explicitly connect the results to the broader field of biopesticide research. Address potential limitations of the study and suggest future research directions. Emphasize the novelty and potential impact of the findings more strongly.
The manuscript should more clearly state the novelty of using CPPs to enhance the efficacy of biopesticides. I recommend that this point should be emphasized in the abstract, introduction, and conclusion. Also, it will be very useful to compare and contrast the presented experiments with existing methods in the discussion to underscore its innovation.
Discuss the broader implications of the findings for the biopesticide industry, including potential environmental benefits and commercial applications. Address how this method could be applied to other pest species or in different agricultural settings.
Ensure all figures and tables are high quality and clearly labelled. The legends should provide sufficient detail to understand the data without referring back to the main text. Consider adding a summary table that compares the efficacy of HxTx-Hv1h alone, HxTx-Hv1h/GNA, and HxTx-Hv1h/CPP-1838. Clearly indicate which details are provided in the supplementary material and ensure it is accessible and well-organized.
Comments on the Quality of English Language
English language comments:
Correct subject-verb agreement errors. For example, "Aphids immersed in buffer solutions without any recombinant protein exhibited survival rates exceeding 80%" should be "Aphids immersed in buffer solutions without any recombinant protein exhibited survival rates exceeding 80%." Also, overly complex sentences should be avoided. For instance, "The inclusion of surfactants, such as Silwet L-77, has been reported to amplify the topical toxicity of HxTx-Hv1h against Drosophila suzukii and can enhance cuticle penetration" could be rephrased as "Surfactants like Silwet L-77 have been shown to amplify the topical toxicity of HxTx-Hv1h against Drosophila suzukii by enhancing cuticle penetration."
I recommend to replace vague terms with specific scientific terminology where appropriate. For example, "The apex of expression was observed in clones containing ten copies of the gene" could be "The highest expression levels were observed in clones containing ten copies of the gene."
Use consistent terminology throughout the manuscript. For instance, if "aphidicidal" is used, ensure it's consistently applied rather than alternating with terms like "insecticidal."
Author Response
Dear Reviewer,
Thank you for your your valuable suggestions and professional comments of the manuscript. Based on your insightful comments, you must be an expert with a solid foundation and a strong sense of responsibility. Your expertise has greatly contributed to the improvement of our paper. We have carefully considered the comments and have revised the manuscript accordingly. The comments and detailed responses can be summarized as follows:
Comment 1: The abstract provides a concise summary but could benefit from clearer statements regarding the significance of the findings. Consider highlighting the novelty of using CPPs in the context of biopesticides in the opening sentence to immediately capture the reader's interest.
Response 1: Thank you for your suggestion. Initially, we considered adding 1-2 sentences to emphasize the novelty of using CPPs in the context of biopesticides, but we had to abandon the idea based on the following reasons. First, the journal has a strict word limit for abstracts, allowing only up to 200 words for submission to the system. Second, about ten days before we submitted our manuscript, Jiang et al. published an article in Molecular Plant (Jiang C, et al. Mol Plant. 2024 Jul 1;17(7):987-989. doi: 10.1016/j.molp.2024.05.013.), where they used the cell penetrating peptide TAT to enhance the delivery efficiency of small RNA-based biopesticides. It should be emphasized that our manuscript is the first to report that CPP can enhance the insecticidal activity of protein-based biopesticides.
Comment 2: The introduction is informative but dense. Dividing long paragraphs into shorter ones with clear subheadings (e.g., "Impact of Herbivorous Insects on Agriculture", "Challenges for Chemical Pesticides", "Potential for Biological Pesticides", etc.) may improve readability.
Response 2: This paragraph aims to highlight the urgent need to develop biopesticides beyond Bt, particularly targeting pests with piercing-sucking mouthparts. We attempted to split the longer paragraphs into shorter ones with clear subheadings, only to find that each paragraph consisted of merely one or two sentences, and suitable subheadings for the following sections of the introduction were not apparent. Therefore, we think it is preferable to maintain the original format.
Comment 3: The rationale for selecting HxTx-Hv1h vs CPP-1838 should be highlighted more clearly. Why was CPP-1838 chosen?
Response 3: We have dedicated almost a paragraph in the introduction of our manuscript to explain the selection of CPP-1838; please refer to the fourth paragraph, lines 90-100. The efficacy of CPP delivery can vary depending on the peptide sequence, the type of cargo being delivered, and the target cell type. Our focus on CPP-1838 was the result of reviewing two key pieces of literature. One of them is a study by Ramaker et al., who used a uniform uptake assay to evaluate the ability of 474 CPPs to transport model cargo into HeLa cells, identifying 20 CPP sequences with exceptional delivery efficiency, including CPP-1838 (Ramaker K, et al. Cell penetrating peptides: a comparative transport analysis for 474 sequence motifs. Drug Deliv. 2018 Nov;25(1):928-937. doi: 10.1080/10717544.2018.1458921.). Second, Darif et al. described the absorption and permeation mechanisms of four CPPs, including CPP-1838, in two insect cell lines and dissected midgut tissues (Darif N, et al. Cell penetrating peptides are versatile tools for enhancing multimodal uptake into cells from pest insects. Pestic Biochem Physiol. 2023 Feb; 190:105317. doi: 10.1016/j.pestbp.2022.105317.). They demonstrated the variability in uptake among different CPPs, with CPP-1838 showing a unique ability to penetrate cells through diffusion and endocytosis. It is precisely because of the exceptional delivery capabilities displayed by CPP-1838 in insect cells that we chose CPP-1838 for our study.
Comment 4: The methods section is detailed, but some descriptions of procedures could have been shortened or moved to supplemental materials to focus on the main results. Ask about any new or particularly difficult procedures, as this will help other researchers replicate your work. The results are clearly shown in appropriate figures, but complex sentences need to be simplified for better understanding, e.g., “Vector pGAPZαA containing a single copy of the HxTx-Hv1h/CPP-1838 expression cassette is shown in detail in the supplementary figures. S1A” can be abbreviated to “Details of the pGAPZαA vector containing a single copy of the HxTx-Hv1h/CPP-1838 expression cassette are shown in Supplementary Figure S1A.”
Response 4: In accordance with your suggestions, we have made the necessary simplification of the material section and modified the sentences you gave as examples and other complex sentences.
Comment 5: The discussion should more explicitly connect the results to the broader field of biopesticide research. Address potential limitations of the study and suggest future research directions. Emphasize the novelty and potential impact of the findings more strongly.
Response 5: In my opinion, the research on developing neurotoxic peptides into protein-based bioinsecticides mainly concentrates on the following aspects: firstly, screening insecticidal peptide molecules that can be developed into bioinsecticides. Secondly, carrying out molecular modifications on the screened peptide fragments to further enhance their oral or contact insecticidal activities. Thirdly, improving the recombinant expression level of insecticidal peptides to reduce their production costs. HxTx-Hv1h is one of the few neurotoxic peptides with oral or contact insecticidal activity, however, its potency still needs to be further improved. Previous research primarily utilized lectins, including GNA, as transport "carriers" for neurotoxic peptides to increase their transport efficiency to the site of action. The present study demonstrates that cell-penetrating peptides like CPP-1838 can also act as transport "carriers" for neurotoxic peptides such as HxTx-Hv1h, to enhance their contact aphicidal activities. Based on this suggestion, we have added some content to the first paragraph of the discussion section.
Comment 6: The manuscript should more clearly state the novelty of using CPPs to enhance the efficacy of biopesticides. I recommend that this point should be emphasized in the abstract, introduction, and conclusion. Also, it will be very useful to compare and contrast the presented experiments with existing methods in the discussion to underscore its innovation.
Response 6: In the abstract, we mentioned that "this study is the first to record the CPP fusion strategy, which enhanced the percutaneous aphicidal activity of natural toxins such as HxTx-Hv1h." In the introduction part of the manuscript, we stated that "our research supports the fusion of HxTx-Hv1h with CPP-1838 as a viable strategy to improve insecticidal efficacy." The conclusion has been rewritten, and it states: "This study demonstrates for the first time that the fusion of the neurotoxic peptide HxTx-Hv1h with the cell-penetrating peptide CPP-1838 can significantly enhance its aphicidal effect." In the discussion section, we have discussed the potential mechanism of action and the enhancement of the insecticidal activity of HxTx-Hv1h by CPP-1838 and GNA. I believe that the aforementioned aspects should be able to emphasize the innovation of our study.
Comment 7: Ensure all figures and tables are high quality and clearly labelled. The legends should provide sufficient detail to understand the data without referring back to the main text. Clearly indicate which details are provided in the supplementary material and ensure it is accessible and well-organized.
Response 7: I had submitted the manuscript text, the high-resolution figures, and supplementary figures and table separately when I initially submitted the paper. It seems that during the editorial process the figures might have been automatically converted when the manuscript was sent for review, which might have resulted in a loss of the original pixel quality. And supplementary figures and table are not included in the document. I have now re-inserted the original high-resolution Figure 1 and supplementary figures into the manuscript. Please refer to the revised manuscript where this modification has been made.
Comment 8: Correct subject-verb agreement errors. For example, "Aphids immersed in buffer solutions without any recombinant protein exhibited survival rates exceeding 80%" should be "Aphids immersed in buffer solutions without any recombinant protein exhibited survival rates exceeding 80%." Also, overly complex sentences should be avoided.
Response 9: We carefully checked the entire manuscript to ensure there were no subject-verb agreement errors. Additionally, we made appropriate adjustments to overly complex sentences in the manuscript.
Comment 10: I recommend to replace vague terms with specific scientific terminology where appropriate. For example, "The apex of expression was observed in clones containing ten copies of the gene" could be "The highest expression levels were observed in clones containing ten copies of the gene."
Response 10: Thank you for your reminder. I have corrected the sentence you mentioned and have revisited the entire document to avoid the use of ambiguous terminology.
Comment 11: Use consistent terminology throughout the manuscript. For instance, if "aphidicidal" is used, ensure it's consistently applied rather than alternating with terms like "insecticidal."
Response 11: "Aphidicidal" means capable of killing aphids, whereas "insecticidal" refers to the killing of insects. "Insecticidal" denotes a broader range of targeted insects and is not limited to the scope implied by "aphidicidal." We conducted a thorough review of the entire manuscript to ensure consistent use of terminology throughout the document. The manuscript uses the terms pesticide and insecticide. Upon consulting the related literature, it is known that all insecticides are pesticides, but not all pesticides are insecticides. The term "pesticide" is used to refer to any substance that controls pests, while "insecticide" specifically refers to agents that target insects. Therefore, it seems that both terms can be used interchangeably in the manuscript.
Please let us know if we need make further improvements. Thank you very much.
Sincerely
Reviewer 2 Report
Comments and Suggestions for Authors
The manuscript is well written and the data are well presented.
Minor comments
The resolution of the figures needs to be enhanced. Blurring is observed.
Typos and language errors, for example first sentence of abstract and similar
Conclusion looks too long and unlike a conclusion rewrite to highlight the major conclusions drawn from the study.
The future perspective of this research needs to be discussed
Comments on the Quality of English Language
Minor edits needed
Author Response
Dear reviewer,
Thank you for reviewing our manuscript and providing valuable comments and suggestions. We have addressed your suggestions point by point as follows:
Comment 1: The resolution of the figures needs to be enhanced. Blurring is observed.
Response 1: I had submitted the manuscript text and the high-resolution figures separately when I initially submitted the paper. Unfortunately, it seems that during the editorial process the figures might have been automatically converted when the manuscript was sent for review, which might have resulted in a loss of the original pixel quality. I have manually replaced the unclear Figure 1 in the revised manuscript. And I will upload the high-resolution images to the system along with the second submission. Thank you for your reminder.
Comment 2: Typos and language errors, for example first sentence of abstract and similar
Response 2: We have rewritten the first sentence of the abstract and reviewed the entire text to ensure there are no spelling or language errors.
Comment 3: Conclusion looks too long and unlike a conclusion rewrite to highlight the major conclusions drawn from the study
Response 3: We modified the Conclusion as follows: “In conclusion, this study presents the first evidence that fusion of the neurotoxic peptide HxTx-Hv1h with the cell-penetrating peptide CPP-1838 significantly improves its efficacy against aphids. The synergistic effect with surfactant Silwet L-77 facilitates enhanced cuticular penetration, leading to superior contact aphidicidal performance. Our findings with the BglBrick methodology also indicate that increased gene copy numbers in P. pastoris cells up to ten copies correlate with higher expression levels of HxTx-Hv1h/CPP-1838. High-density fermentation successfully yielded 2.30 g/L of the active fusion protein, underscoring the potential of CPPs to enhance the effectiveness of protein-based biopesticides.”
Comment 4: The future perspective of this research needs to be discussed
Response 4: In order to avoid exaggerated statements about future prospects in the abstract, we have rewritten the last sentence of the abstract. The rewritten sentence is as follows: “This study is the first to document a CPP fusion strategy that enhances the transdermal aphidicidal activity of a natural toxin like HxTx-Hv1h and opens up the possibility of exploring the recombinant production of HxTx-Hv1h/CPP-1838 for potential applications”.
Please let us know if we need make further improvements. Thank you very much.
Reviewer 3 Report
Comments and Suggestions for Authors
The authors showed that the aphidicidal efficiency of HxTx-Hv1h, spider venom-derived neurotoxin peptides, could improve by conjugation with Cell Penetrating Peptide (CPP). They also showed the possibility of large-scale production of this fusion protein using recombinant yeast containing multi-copy expression vectors. The manuscript is well written, and the methods and results are presented clearly. However, the reviewer has a few comments to improve this work.
Major points
1. Figure 1 looks strange. It seems that the black and white colors are inverted, making the legends unreadable. Please replace it with the correct version.
2. I have some concerns about the Western blotting figure,Fig. 1 B and C. The Western blotting bands for fusion proteins HxTx-Hv1h/GNA and HxTx-Hv1h/CPP-1838 appear to be split into two respectively. The main text did not mention the second band; what is it? If it is a degradation product, it seems that the amount of the degradation product is greater than the target protein particulaly in Fig. 1B. It would be better to change the figures or add a satisfactory explanation in the text. For example, Figure 3 shows a Western blot image of recombinant proteins with a single band.
Minor points
1.In lines 195-197, it was mentioned that the fusion protein alone "did not meet practical application standards," but what LD50 is required for it to be commercially viable? It would be helpful if the discussion mentioned whether a sufficient LD50 was obtained when fusion proteins HxTx-Hv1h/GNA or HxTx-Hv1h/CPP-1838 used in combination with surfactant, as described later.
2. In line 250, the author says "without," but in context, isn't it likely a mistake and should be "with"? Sorry if I’m wrong.
Author Response
Dear reviewer,
Thank you for reviewing our manuscript and providing valuable comments and suggestions. Your expertise has greatly contributed to the improvement of our paper. We have addressed your suggestions point by point as follows:
Comment 1: Figure 1 looks strange. It seems that the black and white colors are inverted, making the legends unreadable. Please replace it with the correct version.
Response 1: Regarding the issue with the image quality, I indeed had submitted the manuscript text and the high-resolution figures separately when I initially submitted the paper. Unfortunately, it seems that during the editorial process the figures might have been automatically converted when the manuscript was sent for review, which might have resulted in a loss of the original pixel quality. I have now re-inserted the original high-resolution Figure 1 into the manuscript. Please refer to the revised manuscript where this modification has been made.
Comment 2: I have some concerns about the Western blotting figure,Fig. 1 B and C. The Western blotting bands for fusion proteins HxTx-Hv1h/GNA and HxTx-Hv1h/CPP-1838 appear to be split into two respectively. The main text did not mention the second band; what is it? If it is a degradation product, it seems that the amount of the degradation product is greater than the target protein particulaly in Fig. 1B. It would be better to change the figures or add a satisfactory explanation in the text. For example, Figure 3 shows a Western blot image of recombinant proteins with a single band.
Response 2: Thank you for this valuable suggestion. I believe you are referring to the Western blotting figure, Fig. 1 B and C, which should actually be the immunoblot figures Fig. 1 C and D in the manuscript, right? Regarding the appearance of two bands in the immunoblot with Anti-His antibody for the fusion proteins HxTx-Hv1h/GNA and HxTx-Hv1h/CPP-1838 in Fig. 1 C, our explanation is as follows: Although the recombinant HxTx-Hv1h/GNA and HxTx-Hv1h/CPP-1838 purified by nickel column affinity chromatography have a purity over 90%, there might still be some impurities that can bind to the Anti-His antibodies. Given that Anti-His antibodies do not have high specificity, the presence of additional bands in the Anti-His immunoblot can occur. In Fig. 1 D, we suspect that the recombinant fusion protein HxTx-Hv1h/GNA may have undergone partial degradation, resulting in the appearance of two bands during the Anti-HxTx-Hv1h immunoblot analysis. This hypothesis is based on the fact that the size of the second, smaller band observed in the immunoblot of HxTx-Hv1h/GNA corresponds to the size of the band obtained with recombinant HxTx-Hv1h. In fact, it was due to the presence of additional bands besides the target band in the immunoblot analysis with Anti-HxTx-Hv1h that led us to further purify the recombinant proteins by ion-exchange chromatography before conducting Western blotting, in order to avoid issues like degradation and impurity-related non-specific bands. In response to this suggestion, we have supplemented the revised manuscript accordingly.
Comment 3: In lines 195-197, it was mentioned that the fusion protein alone "did not meet practical application standards," but what LD50 is required for it to be commercially viable? It would be helpful if the discussion mentioned whether a sufficient LD50 was obtained when fusion proteins HxTx-Hv1h/GNA or HxTx-Hv1h/CPP-1838 used in combination with surfactant, as described later.
Response 3: Personally, I believe that there are no strict and quantifiable standards for the practical application of protein-based bioinsecticides. In China, it's inconceivable for farmers to use a product that costs over several times more than conventional chemical insecticides and doesn't provide immediate pest control effects. Yet, in some developed countries, people might be willing to pay a premium for environmentally friendly or organic labels. In fact, what this statement is intended to mean is that while the fusion with CPP-1838 enhanced the contact toxicity of HxTx-Hv1h, the insecticidal activity of HxTx-Hv1h/CPP-1838 without the addition of surfactants is still not likely to reach the potential for application. To address this issue, we have adjusted the phrasing of this statement; please refer to the revised manuscript.
Comment 4: In line 250, the author says "without," but in context, isn't it likely a mistake and should be "with"? Sorry if I’m wrong.
Response 4: Without the addition of surfactants, aphid protein extracts from the HxTx-Hv1h/GNA and HxTx-Hv1h/CPP-1838 treatment groups exhibited more immunoreactive bands. Conversely, with the addition of surfactant, fewer immunoreactive bands were observed in the HxTx-Hv1h/GNA and HxTx-Hv1h/CPP-1838 treatment groups, suggesting that the degradation of HxTx-Hv1h/GNA and HxTx-Hv1h/CPP-1838 within the aphids was reduced following the addition of surfactant. This result support the view that surfactant increase the stability of HxTx-Hv1h/GNA and HxTx-Hv1h/CPP-1838 inside the aphid body.
Please let us know if we need make further improvements. Thank you very much.
Sincerely